# Synthetic Petri Dish: A Novel Surrogate Model for Rapid Architecture Search

## Abstract

Neural Architecture Search (NAS) explores a large space of architectural motifs – a compute-intensive process that often involves ground-truth evaluation of each motif by instantiating it within a large network, and training and evaluating the network with thousands or more data samples. Inspired by how biological motifs such as cells are sometimes extracted from their natural environment and studied in an artificial Petri dish setting, this paper proposes the *Synthetic Petri Dish* model for evaluating architectural motifs. In the Synthetic Petri Dish, architectural motifs are instantiated in very small networks and evaluated using very few *learned* synthetic data samples (to effectively approximate performance in the full problem). The relative performance of motifs in the Synthetic Petri Dish can substitute for their ground-truth performance, thus accelerating the most expensive step of NAS. Unlike other neural network-based prediction models that parse the structure of the motif to *estimate* its performance, the Synthetic Petri Dish predicts motif performance by training *the actual motif* in an artificial setting, thus deriving predictions from its true intrinsic properties. Experiments in this paper demonstrate that the Synthetic Petri Dish can therefore predict the performance of new motifs with significantly higher accuracy, especially when insufficient ground truth data is available. Our hope is that this work can inspire a new research direction in studying the performance of extracted components of models in a synthetic diagnostic setting optimized to provide informative evaluations.

## 1 Introduction

The architecture of deep neural networks (NNs) is critical to their performance. This fact motivates neural architecture search (NAS), wherein the choice of architecture is often framed as an automated search for effective *motifs*, i.e. the design of a repeating recurrent cell or activation function that is repeated often in a larger NN blueprint. However, evaluating a candidate architecture's ground-truth performance in a task of interest depends upon training the architecture to convergence. Complicating efficient search, the performance of an architectural motif nearly always benefits from increased computation (i.e. larger NNs trained with more data). The implication is that the best architectures often require training near the bounds of what computational resources are available, rendering naive NAS (i.e. where each candidate architecture is trained to convergence) exorbitantly expensive.

To reduce the cost of NAS, methods often exploit heuristic *surrogates* of true performance. For example, motif performance can be evaluated after a few epochs of training or with scaled-down architectural blueprints, which is often still expensive (because maintaining reasonable fidelity between ground-truth and surrogate performance precludes aggressive scaling-down of training). Another approach learns models of the search space (e.g. Gaussian processes models used within Bayesian optimization), which improve as more ground-truth models are trained, but cannot generalize well beyond the examples seen. This paper explores whether the computational efficiency of NAS can be improved by creating a new kind of surrogate, one that can benefit from miniaturized training and still generalize beyond the observed distribution of ground-truth evaluations. To do so, we take inspiration from an idea in biology, bringing to machine learning the application of a *Synthetic Petri Dish* microcosm that aims to identify high-performing architectural motifs.

The overall motivation behind "in vitro" (test-tube) experiments in biology is to investigate in a simpler and controlled environment the key factors that explain a phenomenon of interest in a messier

and more complex system. For example, to understand causes of atypical mental development, scientists extract individual neuronal cells taken from brains of those demonstrating typical and atypical behavior and study them in a Petri dish (Adhya et al., 2018). The approach proposed in this paper attempts to algorithmically recreate this kind of scientific process for the purpose of finding better neural network motifs. The main insight is that biological Petri dish experiments often leverage both (1) key aspects of a system's dynamics (e.g. the behavior of a single cell taken from a larger organism) and (2) a human-designed intervention (e.g. a measure of a test imposed on the test-tube). In an analogy to NAS, (1) the dynamics of learning through backpropagation are likely important to understanding the potential of a new architectural motif, and (2) compact synthetic datasets can illuminate an architecture's response to learning. That is, we can use machine learning to *learn* data such that training an architectural motif on the learned data results in performance indicative of the motif's ground-truth performance.

In the proposed approach, motifs are extracted from their *ground-truth evaluation* setting (i.e. from large-scale NNs trained on the full dataset of the underlying domain of interest, e.g. MNIST), instantiated into very small networks (called *motif-networks*), and evaluated on learned synthetic data samples. These synthetic data samples are trained such that the *performance ordering* of motifs in this Petri dish setting (i.e. a miniaturized network trained on a few synthetic data samples) matches their ground-truth performance ordering. Because the *relative* performance of motifs is sufficient to distinguish good motifs from bad ones, the Petri dish evaluations of motifs can be a surrogate for ground-truth evaluations in NAS. Training the Synthetic Petri Dish is also computationally inexpensive, requiring only a few ground-truth evaluations, and once trained it enables extremely rapid evaluations of new motifs.

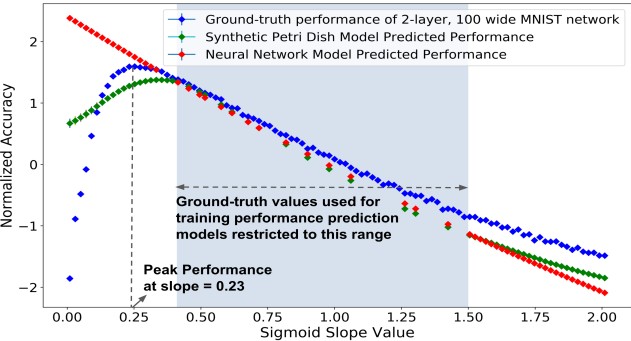

Figure 1: **Predicting the Optimal Slope of the Sigmoid Activation.** Each blue diamond depicts the normalized validation accuracy (ground-truth) of a 2-layer, 100-wide feed-forward network with a unique sigmoid slope value (mean of 20 runs). The validation accuracy peaks at a slope of $0.23$. Both the Synthetic Petri Dish and a neural network surrogate model that predicts performance as a function of sigmoid slope are trained on a limited set of ground-truth points, restricted to the blue-shaded region to the right of the peak. The normalized performance predictions for Synthetic Petri Dish are shown with green diamonds and those for the NN surrogate model are shown as red diamonds. The plot shows that the NN model predictions overfits the training data. In contrast, because the Synthetic Petri Dish conducts experiments with small neural networks with these sigmoid slope values it is more more accurate at inferring both that there is a peak and its approximate location.

A key motivating hypothesis is that because the Synthetic Petri Dish evaluates the motif by actually *using* it in a simple experiment (e.g. training it with SGD and then evaluating it), its predictions can generalize better than other neural network (NN) based models that predict motif performance based on only observing the motif's structure and resulting performance (Liu et al., 2018a; Luo et al., 2018). For example, consider the demonstration problem of predicting the ground-truth performance of a two-layer feedforward MNIST network with sigmoidal non-linearity. The blue points in Figure 1 shows how the ground-truth performance of the MNIST network varies when the slope of its sigmoid activations (the term $c$ in the sigmoid formula $1/(1 + e^{-cx})$) is varied in the range of $0.01 - 2.01$. The MNIST network performance peaks near a slope-value of $0.23$. Similarly to the NN-based model previously developed in Liu et al. (2018a); Luo et al. (2018), one can try to train a neural network that predicts the performance of the corresponding MNIST network given the sigmoid slope value

as input (Section 4.1 provides full details). When training points (tuples of sigmoid slope value and its corresponding MNIST network performance) are restricted to an area to the right of the peak (Figure 1, blue-shaded region), the NN-based prediction model (Figure 1, red diamonds) generalizes poorly to the test points on the left side of the peak ($c < 0.23$). However, unlike such a conventional prediction model, the prediction of the Synthetic Petri Dish generalizes to test points left of the peak (despite their behavior being drastically different than what would be expected solely based on the points in the blue shaded region). That occurs because the Synthetic Petri Dish trains and evaluates the *actual candidate motifs*, rather than just making predictions about their performance based on data from past trials.

Beyond this explanatory experiment, the promise of the Synthetic Petri Dish is further demonstrated on a challenging and compute-intensive language modelling task that serves as a popular NAS benchmark. The main result is that Petri dish obtains highly competitive results even in a limited-compute setting. Interestingly, these results suggest that it is indeed possible to extract a motif from a larger setting and create a controlled setting (through learning synthetic data) where the instrumental factor in the performance of the motif can be isolated and tested quickly, just as scientists use Petri dishes to test specific hypothesis to isolate and understand causal factors in biological systems.

## 2 RELATED WORK

NAS methods have discovered novel architectures that significantly outperform hand-designed solutions (Zoph and Le, 2017; Elsken et al., 2018; Real et al., 2017). These methods commonly explore the architecture search space with either evolutionary algorithms (Suganuma et al., 2017; Miikkulainen et al., 2018; Real et al., 2019; Elsken et al., 2019) or reinforcement learning (Baker et al., 2016; Zoph and Le, 2017). Because running NAS with full ground-truth evaluations can be extremely expensive (i.e. requiring many thousands of GPU hours), more efficient methods have been proposed. For example, instead of evaluating new architectures with full-scale training, heuristic evaluation can leverage training with reduced data (e.g. sub-sampled from the domain of interest) or for fewer epochs (Baker et al., 2017; Klein et al., 2017).

More recent NAS methods such as DARTS (Liu et al., 2018b) and ENAS (Pham et al., 2018) exploit sharing weights across architectures during training to circumvent full ground-truth evaluations. However, a significant drawback of such weight sharing approaches is that they constrain the architecture search space and therefore limit the discovery of novel architectures.

Another approach to accelerate NAS is to train a NN-based performance prediction model that estimates architecture performance based on its structure (Liu et al., 2018a). Building on this idea, Neural Architecture Optimization (NAO) trains a LSTM model to simultaneously predict architecture performance as well as to learn an embedding of architectures. Search is then performed by taking gradient ascent steps in the embedding space to generate better architectures. NAO is used as a baseline for comparison in Experiment 4.2.

Bayesian optimization (BO) based NAS methods have also shown promising results (Kandasamy et al., 2018; Cao et al., 2019). BO models the architecture space using a Gaussian process (GP), although its behavior is sensitive to the choice of a kernel function that models the similarity between any two architectures. Another recent NAS method presents a technique to progressively grow an architecture by adding skip connections, and is named similarly ("Petridish") to the method proposed here (Hu et al., 2019). However unlike the Synthetic Petri Dish introduced here, which is a learned surrogate for NAS, Petridish (Hu et al., 2019) is instead an incremental growth method.

Generative teaching networks (GTNs) also learn synthetic data to accelerate NAS (Such et al., 2020). However, learned data in GTNs helps to more quickly train *full-scale networks* to evaluate their potential on *real* validation data. In the Petri dish, synthetic training and validation instead enables a surrogate microcosm training environment for much smaller extracted motif-networks. Additionally, GTNs are not explicitly trained to differentiate between different networks (or network motifs). In contrast, the Synthetic Petri Dish is optimized to find synthetic input data on which the performance of various architectural motifs is different.

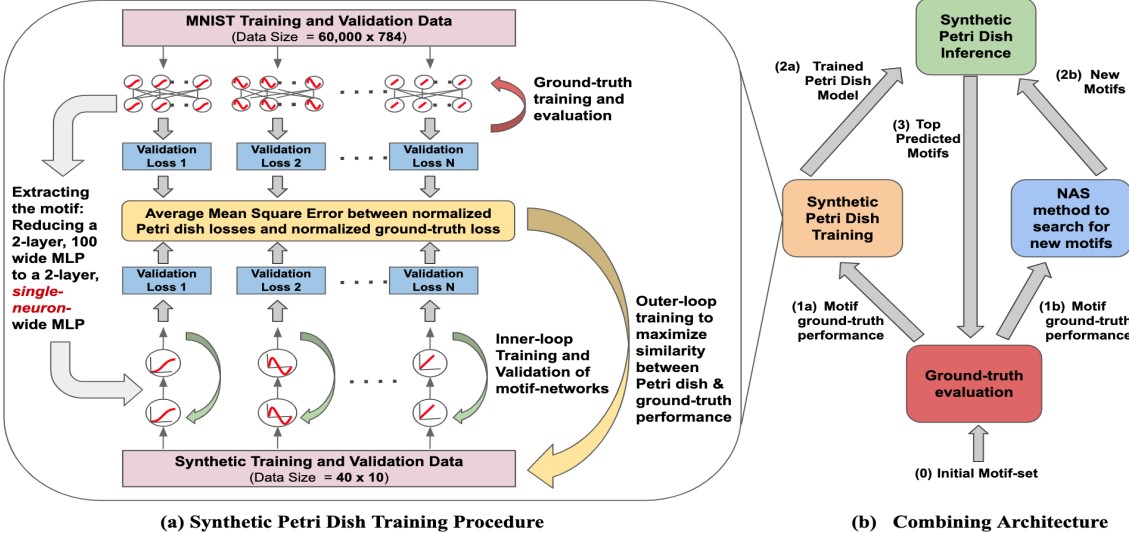

Figure 2: (a) **Synthetic Petri Dish Training.** The left figure illustrates the inner-loop and outer-loop training procedure. The motifs (in this example, activation functions) are extracted from the full network (e.g a 2-layer, 100 wide MLP) and instantiated in separate, much smaller motif-networks (e.g. a two-layer, single-neuron MLP). The motif-networks are trained in the inner-loop with the synthetic training data and evaluated using synthetic validation data. In the outer-loop, an average mean squared error loss is computed between the normalized Petri dish validation losses and the corresponding normalized ground-truth losses. Synthetic training and validation data are optimized by taking gradient steps w.r.t the outer-loop loss. (b) **Combining Architecture Search with the Petri Dish.** Functions are depicted inside rectangles and function outputs are depicted as arrows with their descriptions adjacent to them.

## 3 METHODS

Recall that the aim of the Synthetic Petri Dish is to create a microcosm training environment such that the performance of a small-scale motif trained within it well-predicts performance of the fully-expanded motif in the ground-truth evaluation. First, a few initial ground-truth evaluations of motifs are needed to create training data for the Petri dish. In particular, consider $N$ motifs for which ground-truth validation loss values ($\mathcal{L}_{true}^{i}$, where $i \in 1, 2, ...N$) have already been pre-computed by training each motif in the ground-truth setting. The next section details how these initial evaluations are leveraged to train the Synthetic Petri Dish.

### 3.1 TRAINING THE SYNTHETIC PETRI DISH

To train the Synthetic Petri Dish first requires extracting the $N$ motifs from their ground-truth setting and instantiating each of them in miniature as separate *motif-networks*. For the experiments performed in this paper, the ground-truth network and the motif-network have the same overall blueprint and differ only in the width of their layers. For example, Figure 2a shows a ground-truth network's size reduced from a 2-layer, 100-neuron wide MLP to a motif-network that is a 2-layer MLP with a *single neuron* per layer.

Given such a collection of extracted motif-networks, a small number of *synthetic* training and validation data samples are then learned that can respectively be used to train and evaluate the motif-networks. The learning objective is that the validation loss of motifs trained in the Petri dish resemble the validation loss of the motif's ground-truth evaluation ($\mathcal{L}_{true}^{i}$). Note that this training process requires two nested optimization loops: an inner-loop that trains and evaluates the motif-networks on the synthetic data and an outer-loop that trains the synthetic data itself.

**Initializing the Synthetic Petri Dish:** Before training the Petri dish, the motif-networks and synthetic data must be initialized. Once the motifs have been extracted into separate motif-networks, each motif-network is assigned *the same* initial random weights ($\theta_{init}$). This constraint reduces confound-

ing factors by ensuring that the motif-networks differ from each other *only* in their instantiated motifs. At the start of Synthetic Petri Dish training, synthetic training data ($S^{train} = (x^{train}, y^{train})$) and validation data samples ($S^{valid} = (x^{valid}, y^{valid})$) are randomly initialized. Note that these learned training and validation data can play distinct and complementary roles, e.g. the validation data can learn to test out-of-distribution generalization from a learned training set. Empirically, setting the training and validation data to be the same initially (i.e. $S^{train} = S^{valid}$) benefited optimization at the beginning of outer-loop training; over iterations of outer-loop training, the synthetic training and validation data then diverge. The size of the motif-network and the number of synthetic data samples are chosen through the hyperparameter selection procedure described in Appendix A.2.

**Inner-loop training:** The inner optimization loop is where the performance of motif-networks is evaluated by training each such network independently with synthetic data. This training reveals a sense of the quality of the motifs themselves.

In each inner-loop, the motif-networks are *independently trained* with SGD using the synthetic training data ($S^{train}$). The motif-networks take synthetic training inputs ($x^{train}$) and produce their respective output predictions ($\hat{y}^{train}$). For each motif-network, a binary cross-entropy (BCE) loss is computed between the output predictions ($\hat{y}^{train}$) and the synthetic training labels ($y^{train}$). Because the Petri dish is an artificial setting, the choice of BCE as the inner-loop loss ($\mathcal{L}_{inner}$) is independent of the actual domain loss (used for ground-truth training), and other losses like regression loss could instead be used. The gradients of the BCE loss w.r.t. the motif-network weights inform weight updates (as in regular SGD).

$$\theta_{t+1}^i = \theta_t^i - \alpha \nabla \mathcal{L}_{inner\_train}^i (S^{train}, \theta_t^i) \quad i \in 1, 2, .., N \tag{1}$$

where $\alpha$ is the inner-loop learning rate and $\theta_0^i = \theta_{init}$. Inner-loop training proceeds until individual BCE losses converge. Once trained, each motif-network is independently evaluated using the synthetic validation data ($S^{valid}$) to obtain individual validation loss values ($\mathcal{L}_{inner\_valid}^i$). These inner-loop validation losses then enable calculating an outer-loop loss to optimize the synthetic data, which is described next.

**Outer-loop training:** Recall that an initial sampling of candidate motifs evaluated in the ground-truth setting serve as a training signal for crafting the Petri dish's synthetic data. That is, in the outer loop, synthetic training data is optimized to encourage motif-networks trained upon it to become accurate surrogates for the performance of full networks built with that motif evaluated in the ground-truth setting. The idea is that training motif-networks on the right (small) set of synthetic training data can potentially isolate the key properties of candidate motifs that makes them effective.

To frame the outer-loop loss function, what is desired is for the validation loss of the motif-network to induce the same *relative ordering* as the validation loss of the ground-truth networks; such relative ordering is all that is needed to decide which new motif is likely to be best. One way to design such an outer-loop loss with this property is to penalize differences between normalized loss values in the Petri dish and ground-truth setting[1]. To this end, the motif-network (inner-loop) loss values and their respective ground-truth loss values are first independently normalized to have zero-mean and unit-variance. Then, for each motif, a mean squared error (MSE) loss is computed between the normalized inner-loop validation loss ($\hat{\mathcal{L}}_{inner\_valid}^i$) and the normalized ground-truth validation loss ($\hat{\mathcal{L}}_{true}^i$). The MSE loss is averaged over all the motifs and used to compute a gradient step to improve the synthetic training and validation data.

$$\mathcal{L}_{outer} = \frac{1}{N} \sum_{i=1}^{N} (\hat{\mathcal{L}}_{inner\_valid}^i - \hat{\mathcal{L}}_{true}^i)^2 \tag{2}$$

$$S_{t+1}^{train} = S_t^{train} - \beta \nabla \mathcal{L}_{outer} \quad \text{and} \quad S_{t+1}^{valid} = S_t^{valid} - \beta \nabla \mathcal{L}_{outer} \tag{3}$$

where $\beta$ is the outer-loop learning rate. For simplicity, only the synthetic training ($x^{train}$) and validation ($x^{valid}$) inputs are learned and the corresponding labels ($y^{train}, y^{valid}$) are kept fixed to

---

[1]We tried an explicit rank-loss as well, but the normalized regression loss performed slightly better empirically.

their initial random values throughout training. Minimizing the outer-loop MSE loss ($\mathcal{L}_{outer}$) modifies the synthetic training and validation inputs to maximize the similarity between the motif-networks' performance ordering and motifs' ground-truth ordering.

After each outer-loop training step, the motif-networks are reset to their original initial weights ($\theta_{init}$) and the inner-loop training and evaluation procedure (equation 1) is carried out again. The outer-loop training proceeds until the MSE loss converges, resulting in optimized synthetic data.

### 3.2 PREDICTING PERFORMANCE WITH THE TRAINED PETRI DISH

The Synthetic Petri Dish training procedure described so far results in synthetic training and validation data optimized to sort motif-networks similarly to the ground-truth setting. This section describes how the trained Petri dish can predict the relative performance of unseen motifs, which we call the *Synthetic Petri Dish inference* procedure. In this procedure, new motifs are instantiated in their individual motif-networks, and the motif-networks are trained and evaluated using the optimized synthetic data (with the same hyperparameter settings as in the inner-loop training and evaluation). The relative inner-loop validation loss for the motif-networks then serves as a surrogate for the motifs' relative ground-truth validation loss; as stated earlier, such relative loss values are sufficient to compare the potential of new candidate motifs. Such Petri dish inference is computationally inexpensive because it involves the training and evaluation of very small motif-networks with very few synthetic examples. Accurately predicting the performance ordering of unseen motifs is contingent on the generalization capabilities of the Synthetic Petri Dish (this aspect is further investigated in section 4.1).

### 3.3 COMBINING ARCHITECTURE SEARCH WITH THE SYNTHETIC PETRI DISH

Interestingly, the cheap-to-evaluate surrogate performance prediction given by the trained Petri dish is complementary to most NAS methods that search for motifs, meaning that they can easily be combined. Algorithm 1 in Appendix A.1 shows one possible hybridization of Petri dish and NAS, which is the one we experimentally investigate in this work.

First, the Petri dish model is warm-started by training (inner-loop and outer-loop) using the ground-truth evaluation data ($P_{eval}$) of a small set of randomly-generated motifs ($X_{eval}$). Then in each iteration of NAS, the NAS method generates $M$ new motifs and the Petri dish inference procedure inexpensively predicts their relative performance. The top $K$ motifs (where $K << M$) with highest predicted performance are then selected for ground-truth evaluation. The ground-truth performance of motifs both guides the NAS method as well as provides further data to re-train the Petri dish model. The steps outlined above are repeated until convergence and then the motif with the best ground-truth performance is selected for the final *test evaluation*.

Synthetic Petri Dish training and inference is orders of magnitude faster than ground-truth evaluations, thus making NAS computationally more efficient and faster to run, which can enable finding higher-performing architectures given a limited compute budget.

## 4 EXPERIMENTS

### 4.1 SEARCHING FOR THE OPTIMAL SLOPE FOR SIGMOIDAL ACTIVATION FUNCTIONS

Preliminary experiments demonstrated that when a 2-layer, 100-wide feed-forward network with sigmoidal activation functions is trained on MNIST data, its validation accuracy (holding all else constant) depends on the slope of the sigmoid. The points on the blue curve in Figure 1 demonstrate this fact, where the empirical peak performance is a slope of $0.23$. This simple dependence provides a way to clearly illustrate the benefits of the Synthetic Petri Dish model.

Both the Synthetic Petri Dish and the NN-based surrogate model are trained using 30 ground-truth points that are randomly selected from a restricted interval of sigmoid slope values (the blue-shaded region in Figure 1). The remaining ground-truth points (outside the blue-shaded region) are used only for testing. The NN-based surrogate control is a 2-layer, 10-neuron-wide feedforward network that takes the sigmoid value as input and predicts the corresponding MNIST network validation accuracy as its output. A mean-squared error loss is computed between the predicted accuracy and the ground-truth validation accuracy, and the network is trained with the Adam optimizer.

For training the Synthetic Petri Dish model, each training motif (i.e. sigmoid with a unique slope value) is extracted from the MNIST network and instantiated in a 2-layer, *single-neuron*-wide motif-network ($\theta_{init}$). The setup is similar to the one shown in Figure 2a). The motif-networks are trained in the inner-loop for 200 SGD steps and subsequently their performance on the synthetic validation data ($\mathcal{L}^i_{inner\_valid}$) is used to compute outer-loop MSE loss w.r.t the ground-truth performance (as described in section 3.1). A total of 20 outer-loop training steps are performed. Hyperparameter selection details for the two models are described in Appendix A.3).

Results demonstrate that the NN-based model overfits the training points. (red-curve in Figure 1). In contrast, the Synthetic Petri Dish predictions accurately infer that there is a peak (including the falloff to its left) and also its approximate location (green-curve in Figure 1).

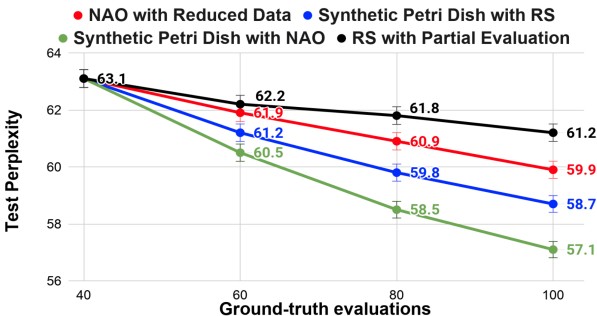

Figure 3: **RNN Cell Search for PTB:** This graph plots the test perplexity (mean of five runs) of the best found cell for four NAS methods across variable numbers of NAS iterations. All the methods are warmed up at the beginning (Step 0 in Figure 2b) with 40 ground-truth evaluations – notice the top-left point with best test perplexity of 63.1. For **Synthetic Petri Dish with RS** (blue curve) and **Synthetic Petri Dish with NAO** (green curve), the top 20 motifs with the best predicted performance are selected for ground-truth evaluations in each NAS iteration. Original NAO (Luo et al., 2018) (not shown here) requires 1,000 ground-truth evaluations to achieve a test perplexity of 56.0. **NAO with Reduced Data** (red-curve) shows the results obtained by running original NAO, but with fewer ground-truth evaluations (the same number the Synthetic Petri Dish variants get). With such limited data, **Synthetic Petri Dish with NAO** outperforms other NAS methods and achieves a test perplexity of 57.1 after 100 ground-truth evaluations.

## 4.2 ARCHITECTURE SEARCH FOR RECURRENT CELLS

The previous experiment demonstrated that a Synthetic Petri Dish model trained with limited ground-truth data can successfully generalize to unseen out-of-distribution motifs. This next experiment tests whether the Synthetic Petri Dish can be applied to a more realistic and challenging setting, that of NAS for a NN language model that is applied to the Penn Tree Bank (PTB) dataset – a popular language modeling and NAS benchmark (Marcus et al., 1993). In this experiment, the architectural motif is the design of a recurrent cell. The recurrent cell search space and its ground-truth evaluation setup is the same as in NAO (Luo et al., 2018). This NAS problem is challenging because the search space is expansive and few solutions perform well (Pham et al., 2018). Each cell in the search space is composed of 12 layers (each with the same layer width) that are connected in a unique pattern. An input embedding layer and an output soft-max layer is added to the cell (each of layer width 850) to obtain a full network (27 Million parameters) for ground-truth evaluation. Each ground-truth evaluation requires training on PTB data-set for 600 epochs and takes 10 hours on a Nvidia 1080Ti.

NAO is one of the state-of-the-art NAS methods for this problem and is therefore used as a baseline in this experiment (called here *original NAO*). In the published results (Luo et al., 2018), *original NAO* requires 1,000 ground-truth evaluations (300 GPU days) over three successive *NAS iterations* to discover a cell with test perplexity of 56.0. These are good results, but the compute cost even to reproduce them is prohibitively large for many researchers. Because the Synthetic Petri Dish offers a potential low-compute option, in this experiment, different NAS methods are compared instead in a setting where only limited ground-truth evaluation data is available ($\leq 100$ samples), giving a sense of how far different methods can get with more reasonable compute resources.

Each *NAS iteration* can be accelerated if the number of costly ground-truth evaluations is reduced by instead cheaply evaluating the majority of candidate motifs (i.e. new cells) in the Petri dish. For the purpose of training the Synthetic Petri Dish, each cell is extracted from its ground-truth setting (850 neurons per layer) and is instantiated in a motif-network with three neurons per layer (its internal cell connectivity, including its depth, remains unchanged). Thus, the ground-truth network that has 27 million parameters is reduced to a motif-network with only 140 parameters. To train motif-networks, synthetic training and validation data each of size $20 \times 10 \times 10$ *(batch size×time steps×motif-network input size)* is learned (thus replacing the 923k training and 73k validation words of PTB). The Petri dish training and inference procedure is very similar to the one described in Experiment 4.1, and it adds negligible compute cost (2 extra hours for training, and a few minutes for inference on a CPU).

Following the steps outlined in algorithm 1 and Figure 2b, the Petri dish surrogate can be combined with two existing NAS methods: (1) Random Search (RS) or (2) NAO itself, resulting in two new methods called Synthetic Petri Dish-RS and Synthetic Petri Dish-NAO. Also, the Random Search NAS method can be combined with partial evaluations resulting in another baseline (Appendix A.4).

For the Synthetic Petri Dish variants, at the beginning of search, both the Petri dish surrogate and the NAS method (RS/NAO) used within the Petri dish variant are warm-started with the ground-truth data of an initial motif set (size 40). In each NAS iteration, 100 newly generated motifs (variable $M$ in algorithm 1) are evaluated using the Petri dish inference procedure and only the top 20 predicted motifs (variable $K$ in algorithm 1) are evaluated for their ground-truth performance. The test perplexity of the best found motif at the end of each NAS iteration is plotted in Figure 3 – the blue curve depicts the result for Synthetic Petri Dish-RS and green depicts the result for Synthetic Petri Dish-NAO. For a fair comparison, original NAO is re-run in this limited ground-truth setting and the resulting performance is depicted by the red-curve in Figure 3. The results show that Synthetic Petri Dish-NAO outperforms both Synthetic Petri Dish-RS and NAO when keeping the amount of ground-truth data points the same, suggesting that the Synthetic Petri Dish and NAO complement each other well. The hybridization of Synthetic Petri Dish and NAO finds a cell that is competitive in its performance (test perplexity 57.1) with original NAO (56.0), using only $1/10^{th}$ of original NAO's compute (and exceeds the performance of original NAO when both are given equivalent compute).

## 5 DISCUSSION AND CONCLUSIONS

In the general practice of science, often the question arises of what factor accounts for an observed phenomenon. In the real world, with all its intricacy and complexity, it can be difficult to test or even formulate a clear hypothesis on the relevant factor involved. For that reason, often a hypothesis is formulated and tested in a simplified environment where the relevant factor can be isolated from the confounding complexity of the world around it. Then, in that simplified setting it becomes possible to run rapid and exhaustive tests, as long as there is an expectation that their outcome might correlate to the real world. In this way, the Synthetic Petri Dish is a kind of microcosm of a facet of the scientific method, and its synthetic data is the treatment whose optimization tethers the dynamics within the simplified setting to their relevant counterparts in the real world.

By approaching architecture search in this way as a kind of question-answering problem on how certain motifs or factors impact final results, we gain the intriguing advantage that the prediction model is no longer a black box. Instead, it actually contains within it a critical piece of the larger world that it seeks to predict. This piece, a motif cut from the ground-truth network (and its corresponding learning dynamics), carries with it from the start a set of priors that no black box learned model could carry on its own. These priors pop out dramatically in the simple sigmoid slope experiment – the notion that there is an optimal slope for training and roughly where it lies emerges automatically from the fact that the *sigmoid slope itself* is part of the Petri Dish prediction model. In the later NAS for recurrent cells, the benefit in a more complex domain also becomes apparent, where the advantage of the intrinsic prior enables the Petri Dish to have better performance than a leading NAS method when holding the number of ground-truth evaluations constant, and achieves roughly the same performance with 1/10th the compute when allowing differing numbers of ground-truth evaluations.

It is also possible that other methods can be built in the future on the idea of extracting a component of a candidate architecture and testing it in another setting. The opportunity to tease out the underlying causal factors of performance is a novel research direction that may ultimately teach us new lessons on architecture by exposing the most important dimensions of variation through a principled empirical process that could capture the spirit and power of the scientific process itself.

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
