# OpenReview forum: "Synthetic Petri Dish: A Novel Surrogate Model for Rapid Architecture Search"
_ICLR.cc/2021/Conference — Reject_

### Official Review · AnonReviewer2 · 2020-10-27
**Many shortcomings, and not yet ready for publication**

**Rating:** 4
**Confidence:** 4

**Review:**

This paper presents an approach to accelerating NAS with 'petri-dish' networks, which hope to mimic the response of original networks at a fraction of training time cost. The key idea is to evaluate an architectural setting on a miniaturized network as opposed to the original network. With this approach computational effort is saved by eschewing expensive 'ground truth' original network evaluations.

The largest hinderance by far in the paper is the quality of the writing. Much of the paper is not well written, and difficult to understand. In particular, the design section is written almost completely in prose without a well delineated algorithm. The paper does not do a good job of guiding the reader through their proposed approach, thus much hunting and guessing is required to understand exactly what it is that the authors are proposing. E.g., at first glance I believed that once the 'petri-dish' was initialized then ground truth evaluations would not be carried out again as it seems that is what the introduction is implying. However the authors combine the petri-dish with ground truth evaluations (Sec. 3.3) but this is not clearly presented in the introduction. Another difficulty is it's unclear exactly the precise experimental settings used, making it impossible to be sure whether the experiments are 'fair' to competing approaches. E.g., to generate Fig. 1 a 'petri-dish' is initialized and trained, however is it true that 'inference' carried out using this petri dish involves training a petri-dish network to convergence then evaluating its accuracy on the synthetic validation data? Although 'petri-dish inference' is defined in Sec 3.2, I see no usage of this exact verb in Sec 4.1. It's difficult to connect these concepts as a reader without a well delineated algorithm which concretizes with precision what is being proposed.

Besides these issues, the proposed idea is not fully well explored which also leads to doubt on the robustness of the approach. One concern I have is that the robustness of petri dish idea at mimicking the original network is not well established outside of a rather simple experiment in Fig. 1. As this is a key linchpin to the proposed approach, I strongly require an extensive experiment in a complex setting (with multiple properties of a network being searched over). Another concern with Fig. 1 is that the 'ground-truth' evaluations used to seed the two models appear to be artificially restricted to the shaded box. Neural networks operate under the i.i.d. setting thus making this approach biased against the NN Model approach. The training and test dataset on this experiments should be sampled i.i.d. unless the authors can provide a compelling reason why.

With regards to Fig. 3, from what I can understand the authors considered usage of NAO over 3 iterations of 33 motifs each to yield 100 ground truth evaluation. Did the authors consider 5 iterations of NAO with 20 motifs each as in the petri-dish approach. What do the authors mean exactly with, "For a fair comparison, original NAO is re-run in this limited ground-truth setting and
the resulting performance is depicted by the red-curve in Figure 3"? Can the authors exactly delineate their experimental settings in the appendix or in the main text.

I note that evaluation was not performed for CNNs. Also it would be nice to see the performance of this approach on standard NAS benchmarks such as NASBench.


Another shortcoming with the proposed approach is that the authors haven't proposed a 'general purpose' method to create petri dishes, and only width reduction is explored. Although this is a nice start, clearly this will not work if the architecture parameter being optimized over using NAS is 'width' itself. Have the authors considered such a scenario? How do the authors propose to create petri-dishes in the generalized setting? This shortcoming is glaring, and I'm not sure whether the proposed idea is well explored with a general purpose 'petri-dish' creation mechanism.

Pros:
-Possibly neat idea if executed well.

Cons:
-See above.

---

> ### Author Response · Authors · 2020-11-25
> **Part 1 of Response to AnonReviewer2**
>
> $\Vert$  the design section is written almost completely in prose without a well delineated algorithm.
>
> We did summarize our method in Algorithm 1, but had to move it to the Appendix due to space constraints. This algorithm captures various aspects of Synthetic Petri Dish including initialization, training, inference and hybridization with other NAS methods.
>
> $\Vert$   At first glance I believed that once the 'petri-dish' was initialized then ground truth evaluations would not be carried out again as it seems that is what the introduction is implying. However the authors combine the petri-dish with ground truth evaluations (Sec. 3.3).
>
> Once the Petri Dish is warm-started with few ground-truth evaluations, it need not be re-trained with fresh ground-truth evaluations. For example, in Experiment 1 (search for the optimal slope of sigmoid), we train the Petri Dish only once and then utilize it to predict the performance of test points.
>
> In Experiment 2, we combine Synthetic Petri Dish with popular NAS methods such as NAO (Luo et al., 2018) and Random Search. These NAS methods conduct their search in an iterative manner where each iteration involves ground-truth evaluation of the newly generated architectures. Since training the Petri Dish is cheap (requires just 2 hours on a CPU), we utilize the fresh ground-truth data from each NAS iteration to re-train the Petri Dish as well. This combined procedure of NAS with Petri Dish is explained in Figure 2 (b) and also in Algorithm 1.
>
> $\Vert$  is it true that 'inference' carried out using this petri dish involves training a petri-dish network to convergence then evaluating its accuracy on the synthetic validation data?
>
> Yes, that is correct. We also mention this detail in the paper. For example, section 3.1 states that -- "Inner-loop training proceeds until individual BCE losses converge. Once trained, each motif-network is independently evaluated using the synthetic validation data to obtain individual validation loss values". Subsequently, Section 3.2 states that during Petri Dish inference procedure, "motif-networks are trained and evaluated using the optimized synthetic data with the same hyper-parameter settings as in the inner-loop training and evaluation."
>
> $\Vert$  Although 'petri-dish inference' is defined in Sec 3.2, I see no usage of this exact verb in Sec 4.1
>
> Section 4.2 refers to Algorithm 1 and Figure 2 (b), both of which highlight the 'Petri Dish Inference' procedure.
>
> $\Vert$  One concern I have is that the robustness of petri dish idea at mimicking the original network is not well established outside of a rather simple experiment in Fig. 1.
>
> Based on your feedback, we are currently working to add results from running Petri Dish for NASBench  in our paper. This will be included in our next revision very soon.
>
> $\Vert$  Another concern with Fig. 1 is that the 'ground-truth' evaluations used to seed the two models appear to be artificially restricted to the shaded box. The training and test dataset on this experiment should be sampled i.i.d. unless the authors can provide a compelling reason why.
>
> We restricted the training points for the performance prediction models to the blue-shaded region so that we can effectively test their generalization capability for unseen points. While training the NN model, we randomly sample batches of ground-truth points from the blue-shaded region. The experimental details are further described in Appendix A.3.
>
> $\Vert$  With regards to Fig. 3, from what I can understand the authors considered usage of NAO over 3 iterations of 33 motifs each to yield 100 ground truth evaluation. Did the authors consider 5 iterations of NAO with 20 motifs each as in the petri-dish approach. What do the authors mean exactly with, "For a fair comparison, original NAO is re-run in this limited ground-truth setting and the resulting performance is depicted by the red-curve in Figure 3"? Can the authors exactly delineate their experimental settings in the appendix or in the main text.
>
> We believe the reviewer has misunderstood our experimental setup for Experiment 2. All NAS variants in Experiment 2 (including NAO with Reduced Data) are run for the same number of iterations. Section 4.2 of the paper describes this in detail. For example, it states that -- "In each NAS iteration, 100 newly generated motifs (variable M in algorithm 1) are evaluated using the Petri dish inference procedure and only the top 20 predicted motifs (variable K in algorithm 1) are evaluated for their ground-truth performance. The test perplexity of the best found motif at the end of each NAS iteration is plotted in Figure 3".
>
> We will try to further clarify this point in our revision.

---

> > ### Author Response · Authors · 2020-11-25
> > **Part 2 of Response to AnonReviewer2**
> >
> > $\Vert$  Another shortcoming with the proposed approach is that the authors haven't proposed a 'general purpose' method to create petri dishes, and only width reduction is explored. Although this is a nice start, clearly this will not work if the architecture parameter being optimized over using NAS is 'width' itself. Have the authors considered such a scenario?
> >
> > This is a very good point. We have demonstrated the effectiveness of our approach for searching activation functions and connectivity of cell microarchitectures. Modifying the Petri Dish to search over the width of the network is an interesting future-work direction. We have some initial ideas in this direction -- for example, architectures with variable layer-widths could be reduced in size by a fixed factor to create their corresponding motif-networks with variable widths.

---

### Official Review · AnonReviewer1 · 2020-10-27
**Very nice idea, but important details unclear**

**Rating:** 6
**Confidence:** 3

**Review:**

tHe paper proposes a method for quickly and cheaply determining the value of a
particular motif in neural architecture search by isolating it in a "petri dish"
that allows it to be evaluated without having to train an entire network. The
authors describe their method and evaluate it empirically, showing the promise
of the method.

The idea is interesting and seems very promising. As the authors say, it
addresses one of the bottlenecks in neural architecture search -- the presented
research deals with an important problem. There are clear reasons for preferring
the proposed method over alternatives.

However, there are important details missing. In particular, it was unclear to
me after reading the paper how the synthetic data are generated and how exactly
the motifs are instantiated. These are crucial parts of the proposed method, and
its success hinges on representative data and embeddings. This should be
explained in some detail; in particular because the paper leaves the reader at a
loss on how to apply this methodology to their own problems.

Further, the evaluation performed by the authors, albeit showing good results,
is very small. Only anecdotal results are presented in two contexts, and it is
unclear how the approach actually performs -- how good are the predictions the
petri dish makes, and how does the derived ranking compare to the ground-truth
ranking?

In summary, while I feel that the idea is interesting and promising, there is
insufficient information for it to have a significant impact.

---

> ### Author Response · Authors · 2020-11-25
> **Response to AnonReviewer1**
>
> Thank you recognizing the promising idea presented in this paper.
>
> $\Vert$   how the synthetic data are generated and how exactly the motifs are instantiated?
>
> We would like to clarify that the synthetic training and validation data is not generated. Instead, the synthetic data is randomly initialized at the start of the experiment and subsequently optimized during each outer-loop iteration (as detailed in Equations (2) and (3)). It is also possible to train a generator for the synthetic data and that could be an interesting direction for future-work.
>
> Each motif variant is extracted from the ground-truth network and instantiated in the corresponding motif-network by reducing the width of the motif, while ensuring that the depth and the internal connectivity of the motif remains unchanged. For example, in the RNN cell search experiment, the ground-truth network has a cell that is composed of 12 layers (layer width of 850) and has millions of parameters. Each cell is extracted from its ground-truth setting and is instantiated in the motif-network with three neurons per layer.
>
> The width of the motif-network and the number of synthetic data samples are chosen through the hyper-parameter selection procedure described in Appendix A.2.
>
> $\Vert$ how good are the predictions the petri dish makes, and how does the derived ranking compare to the ground-truth ranking?
>
> One of the goals of Experiment 1 in the paper (sigmoid slope search) is to compare the performance ordering (i.e. ranking) of Petri Dish predictions with the ground-truth performance ordering.
>
> Also, based on your feedback, we are currently working to add results from another benchmark in our paper. This will be included in our next revision very soon.

---

### Official Review · AnonReviewer4 · 2020-10-28
**Interesting idea**

**Rating:** 6
**Confidence:** 1

**Review:**

##########################################################################

Summary:


The paper provides a novel surrogate model method for Neural Architecture Search. Authors motivated the paper from biological study of cells in an artificial petri dish setting. Authors explains the concept of evaluation of small motifs with synthetic learned training and validation data that can predict the performance of larger network.
Leveraging this, authors proposes speeding up the problem of Neural Architecture Search.

##########################################################################

Pros:

1. The problem of Neural Architecture Search is an impactful problem specially with the recent advances in neural networks. This paper tries to solve the scalability issue of this problem which I think has real world impact.

2. The paper is generally well motivated. The motivation behind the methods are well explained. I liked the petri dish inspiration. Also I think the idea of motif for architecture search deserves more attention. I like the use of motifs to speed/scale up the search procedure such that we can train on smaller set of motifs but do inference on a larger set. Also as the motifs are directly evaluated rather than using some NN to evaluate them, it makes the case for their method more convincing.

3. Overall the paper is well written.  Result section is also well structured.


##########################################################################

Cons:

1. Although the paper puts a lot of importance on motifs, but it does not explain a standard way to generate the motifs for any kind of networks, which makes the scope a bit narrow. Is there a way to standardize the motif generation procedure for different architectures?

2. The performance improvement with time over other methods is not properly shown in the result section. I thought that would have shown more impact in the result. Although it shows perplexity with ground truth evaluations curve, but time vs performance could have probably shown the effect of cheap inference for motifs in these work.

3. Result for only one task is shown in the result section. It's not clear if this method can be for other tasks? Adding more tasks might have strengthened the paper.

4. Authors mentions that original NAO reaches 56 perplexity after 300 GPU days. How many days does it take for the synthetic petri dish with NAO to reach this result? What is its best performance? When does it converges if we keep running for a long time? What is the performance for RS without NAO if we keep running for a long time?

##########################################################################

Questions during rebuttal period:


Please address and clarify the cons above


#########################################################################

---

> ### Author Response · Authors · 2020-11-25
> **Response to AnonReviewer4**
>
> Thank you for recognizing the novelty of the method and the clarity of our paper.
>
> $\Vert$  Is there a way to standardize the motif generation procedure for different architectures?
>
> Each motif variant is extracted from the ground-truth network and instantiated in the corresponding motif-network by reducing the width of the motif, while ensuring that the depth and the internal connectivity of the motif remains unchanged. For example, in the RNN cell search experiment, the ground-truth network has a cell that is composed of 12 layers (layer width of 850) and has millions of parameters. Each cell is extracted from its ground-truth setting and is instantiated in the motif-network with just three neurons per layer. The width of the motif-network is a hyper-parameter that is selected using the procedure described in Appendix A.2.
>
> The same motif-network generation procedure can be used for searching the connectivity of macro-architectures as well.
>
> $\Vert$ Result for only one task is shown in the result section. It's not clear if this method can be for other tasks? Adding more tasks might have strengthened the paper.
>
> Based on your feedback, we are currently working to add results from another benchmark in our paper. This will be included in our next revision very soon.
>
>
> $\Vert$ Authors mentions that original NAO reaches 56 perplexity after 300 GPU days. How many days does it take for the synthetic petri dish with NAO to reach this result?
>
> In the experiments performed in the paper, our goal was to evaluate the performance-prediction mechanism of Synthetic Petri Dish with limited compute resources. For a fair comparison, we also re-run the original NAO code in the same setting and report the numbers in the paper. Running these experiments longer becomes prohibitively expensive, especially given that each of them needs to be run five times with random seed for statistical significance.

---

### Official Review · AnonReviewer3 · 2020-10-28

**Rating:** 6
**Confidence:** 3

**Review:**

Summary:
The paper considers the problem of Neural Architectural Search, and proposes an efficient method called Synthetic Petri Dish inspired by how in-vitro experiments are done in biology. Numerical simulations are shown using two examples to demonstrate the validity of the proposed method.

Pros:
- The paper introduces a nice idea to speed up the problem of architecture/hyper-parameter search. The current search methods are often too prohibitive computationally, so this is a useful contribution.
- The idea is well-motivated, and also quite general in terms of its applicability. As in, this method could be used to speed up a variety of architecture/hyper-parameter searches.

Cons:
- The numerical evidence provided in the paper is quite minimal. It would be very interesting to see the performance of the method across different datasets as well as across different hyper-parameters. For instance, how does the method perform when it comes to choice of layers in a CNN (like different kinds of residual blocks) or transformers, learning rates, number of filters, etc. While the evidence provided in the paper is alright, there is lots of room to strengthen the paper.

Comments:
- While I understand why Figure 1 is introduced very early in the paper, I feel it should be explained better. In particular, the use of normalised validation accuracy without defining it properly could lead to confusion. Is it possible to just use standard metrics like accuracy instead?
- Equation (2): I do not understand why one has to define ground-truth loss and motif network loss. Because the two are subtracted, wouldn't the result be just the difference between the ground truth prediction and motif network prediction? In other words, (a-b)-(c-b) = (a-c). So essentially you are trying to minimise the difference between the predictions of the two models right? Is this also why you only optimise on the synthetic input and not bother about the synthetic output?

---

> ### Author Response · Authors · 2020-11-25
> **Response to AnonReviewer3**
>
> Thank you for recognizing the applicability of Petri Dish to speed up a variety of hyper-parameter searches.
>
> $\Vert$  Is it possible to just use standard metrics like accuracy instead for Figure 1?
>
> We plot the normalized accuracy instead of the raw accuracy in Figure 1 because the raw ground-truth accuracy values need not match the raw motif-network accuracy values. Since the goal of Synthetic Petri Dish is to predict the performance ordering of motifs, we normalize the motif-performance values before plotting them.
>
> $\Vert$  I do not understand why one has to define ground-truth loss and motif network loss.
>
> We would like to clarify that we do not subtract the raw ground-truth loss and the raw motif-network loss. Instead, as explained in Section 3.1 and Equation (2), we construct the outer-loop loss by taking the difference between the normalized ground-truth loss and the normalized motif-network loss. The outer-loop loss tries to minimize the difference in the performance ordering of the motif networks and the ground-truth networks. To frame the outer-loop loss function, what is desired is for the validation loss of the motif-network to induce the same relative ordering as the validation loss of the ground-truth networks; such relative ordering is all that is needed to decide which new motif is likely to be best. One way to design such an outer-loop loss with this property is to penalize differences between normalized loss values in the Petri dish and ground-truth setting. To this end, the motif-network (inner-loop) loss values and their respective ground-truth loss values are first independently normalized to have zero-mean and unit-variance. Then, for each motif, a mean squared error (MSE) loss is computed between the normalized inner-loop validation loss and the normalized ground-truth validation loss. The MSE loss is averaged over all the motifs and used to compute a gradient step to improve the synthetic training and validation data.

---

### Author Response · Authors · 2020-11-25
**Response to all the reviewers**

We would like to thank all reviewers for their insightful comments! We recognize that reviewing is time-consuming work, and we are deeply appreciative. We are glad that the reviewers found 'Synthetic Petri Dish' to be a novel and well motivated method with a potential of real world impact. Below, we’ve written responses to each reviewer individually.

---

### Decision · Program_Chairs · 2021-01-07
**Final Decision**

**Decision:**

Reject

**Comment:**

The reviewers overall appreciated the efforts of the authors in making NAS more computationally efficient. The paper could greatly benefit from further editing/restructuring with the goal of improving clarity, as it’s currently hard to navigate and understand in places. Future submissions of this work would benefit from more extensive empirical validation that motif networks mimic the original network. The reviewers also agreed that for the method to be appealing/useful, a general way to generate motif networks is needed. Overall, the outcome was that this is a very interesting idea but needs further development along the directions outlined above.